# UVB-/Age-Dependent Upregulation of Inflammatory Factor Interleukin-6 Receptor (IL-6R) in Keratinocytes Stimulates Melanocyte Dendricity

**DOI:** 10.3390/ijms262210971

**Published:** 2025-11-12

**Authors:** Daigo Inoue, Koji Ohba, Takako Shibata

**Affiliations:** MIRAI Technology Institute, Shiseido Co., Ltd., 1-2-11 Takashima, Nishi-ku, Yokohama 220-0011, Kanagawa, Japan; koji.oba@shiseido.com (K.O.); takako.shibata@shiseido.com (T.S.)

**Keywords:** solar lentigo, keratinocyte, melanocyte, dendrite, Interleukin-6 receptor, UVB

## Abstract

Ultraviolet (UV) irradiation stimulates melanogenesis in melanocytes and melanin transfer to keratinocytes, where the former is mediated by pleiotropic factors such as SCF, α-MSH, and endothelin-1 (ET-1) secreted by keratinocytes. Therefore, the interaction between melanocytes and keratinocytes after UVB exposure appears to be critical to stimulating melanogenesis. The factors that are responsible for inflammation, one of the key biological processes, are crucial to forming the chronic inflammatory microenvironment in solar lentigines (hereafter called age spots). While chronic inflammation is thought to be involved in hyperpigmentation, the molecular mechanisms through which microinflammation affects melanocyte activation in age spots have not been elucidated. In our study, immunohistochemical analysis showed that the expression of the inflammatory factor IL-6R is enhanced in age spots. Specifically, in cultured keratinocytes irradiated with 10 mJ/cm^2^ UVB, the expression of IL-6R was upregulated in UVB exposure- and age-dependent manners, and the co-culture of melanocytes with UVB-irradiated keratinocytes further demonstrated that melanocyte dendrites increased in length and number in a keratinocyte-age-dependent manner. Moreover, the suppression of IL-6R function in keratinocytes by an IL-6R-specific neutralizing antibody, Tocilizumab, inhibited melanocyte dendricity. These results indicate that the age- and UVB-dependent upregulation of IL-6R in keratinocytes stimulates melanocyte dendricity, which may also contribute to excessive melanin deposition in age spots.

## 1. Introduction

Age spots, among the most conspicuous aging signs, are characterized by excessive production and deposition of melanin pigment in basal keratinocytes in the epidermis; this is mainly due to genetic, environmental, and lifestyle factors [1] but can also be ascribed to external factors such as exposure to UVB and particle matter 2.5 (PM2.5) [2]. Additionally, the increase in number and the darkening of age spots are positively correlated with advancing age according to epidemiological investigation [3]. While age spots do not directly cause health issues, they represent a long-standing esthetic concern and have been actively studied as a pigmentary disorder model in cosmetic and pigment cell research. Mechanistic studies on age spots have revealed not only melanocytic but also age spot-specific aging processes: senescence of keratinocytes and fibroblasts [4,5], chronic inflammation of both the epidermis and dermis layers [6], and dysfunctional differentiation of keratinocytes with metabolic and proliferative reductions [7]. The age spot-specific microenvironment is closely linked to changes in the cytokine and hormone secretory patterns of epidermal cells, which underlie the stimulation of melanocytes for melanin synthesis and, vice versa, the transfer of melanin to keratinocytes [8]. However, the factors that contribute to chronic microinflammation in age spots by stimulating melanogenesis and melanin transfer have not yet been determined. In our previous study, we reported that the downregulation of E-cadherin protein expression in the epidermis of age spots resulted in the upregulation of inflammatory factors such as IL-1α and IL-6 in keratinocytes [9]. Especially, IL-6 and its receptor, IL-6R, are involved in chronic inflammation in other tissues, but their function in age spots has not been determined. Here, we focus on the soluble form of IL-6R (sIL-6R) cleaved from membrane-bound IL-6R, which is associated with the enhancement in the proinflammatory responses in skin, including psoriasis atopic dermatitis and barrier repair following injury, triggered by the signal “alarmin” [10]. We found that the release of sIL-6R by keratinocytes is upregulated in UVB exposure- and age-dependent manners and that the increase in the length and number of dendrites induced in melanocytes by this protein is age-dependent. In age spots, the expression of the IL-6R protein, possibly in its soluble form, was upregulated in the entire epidermis. Thus, these results indicate that the sIL-6R-mediated stimulation of melanocyte dendricity is a crucial step in the hyperpigmentation and chronic inflammation of age spots.

## 2. Results and Discussion

We first aimed to determine whether there was a change in the expression of sIL-6R secreted by keratinocytes after UVB exposure in an age-dependent manner. To this end, the levels of the sIL-6R protein in the culture medium of keratinocytes from young (ages 0 and 17) and adult (ages 48, 56, and 57) donors were compared. The basal expression level of sIL-6R in the adult keratinocytes was significantly higher than that in the young cells before UVB exposure (Figure 1A). Additionally, UVB irradiation significantly increased sIL-6R expression in the adult keratinocytes compared with the young cells, with an approximately 2-fold difference (Figure 1A). These results indicate that the release of sIL-6R is increased in both UVB exposure- and age-dependent manners. Although the IL-6 signaling pathway is known to be involved in the proliferation and differentiation of keratinocytes [11], the function of sIL-6R itself has not been sufficiently elucidated in both keratinocytes and melanocytes. Since the downstream signaling cascade of IL-6R is a JAK-STAT3 pathway, we next examined whether STAT3 was activated by detecting phospho-Tyr705 of STAT3 in the co-culture of UVB-irradiated keratinocytes and normal melanocytes. Although we observed an increase in the release of sIL-6R in the keratinocyte-only culture, these cells hardly showed any nuclear-localized pY705-STAT3 in the co-culture (Figure 1B, black arrowheads). On the other hand, the melanocytes co-cultured with both young and adult keratinocytes did show nuclear localized pY705-STAT3 (Figure 1B, white arrowheads), with no significant difference among co-cultures with different ratios adult and young keratinocytes (Figure 1B). Additionally, neither the number nor the melanin content in melanocytes exhibited significant differences between adult and young keratinocyte co-cultures (Figure 1C). These results indicate that, after UVB exposure, sIL-6R from keratinocytes activates the JAK-STAT3 pathway in melanocytes but does not affect melanocyte proliferation and melanogenesis. Based on the above results, we further examined whether upregulated sIL-6R had any effect on dendrite formation in melanocytes, which is crucial to the stimulation of melanin transfer to keratinocytes after UVB exposure [12]. Melanocytes co-cultured with adult UVB-treated keratinocytes exhibited a significant increase in both the number and length of dendrites, to a greater extent than those co-cultured with young UVB-irradiated keratinocytes (Figure 2A). To test whether such enhancement in melanocyte dendricity was dependent on the function of sIL-6R, we further utilized an IL-6R-specific neutralizing antibody, Tocilizumab, to suppress this receptor [13]. UVB-irradiated keratinocytes were treated with either Tocilizumab or control IgG for 24 h, which was then rinsed off, and were subsequently co-cultured with melanocytes (Figure 2B). The melanocytes co-cultured with UVB/Tocilizumab-treated keratinocytes exhibited fewer and shorter dendrites than the control counterparts (UVB/control IgG-treatment; Figure 2B), which indicates that the upregulation of melanocyte dendricity is dependent on IL-6R function. We next examined whether the expression of IL-6R in age spots differs from that in normal epidermis. For a homogeneous comparison, we compared the fluorescence intensity of the perilesional and lesional areas in the same section based on fluorescent immunostaining and found that the expression of the IL-6R protein was upregulated in the basal, stratum, and granular layers of age spots but not in the neighboring non-lesion area (Figure 2C). Therefore, in age spots, the upregulation of IL-6R may be involved in the stimulation of melanocyte dendricity, which may in turn contribute to transferring melanin to basal keratinocytes in excessive quantities.

Our study revealed that the age-related alteration in the keratinocyte response to UVB exposure is due to the upregulated release of sIL-6R by adult keratinocyte cells, which in turn induces an increase in the number and length of melanocyte dendrites. These findings indicate that not only UVB but also the aging process could be crucial to forming an inflammatory microenvironment, thereby inducing melanogenesis in melanocytes and melanin transfer to keratinocytes. IL-6 is a senescence-related cytokine secreted in the early stages of skin tissue damage due to wounds or UV irradiation [14]. Furthermore, it is also well known that IL-6/IL-6R is crucial for chronic inflammatory disorders such as rheumatoid arthritis, psoriasis, and vitiligo, in which ratio of IL-6/IL-6R and expressional timing of both IL-6 and IL-6R via cis-, trans, and cluster signaling seem to be crucial for the pleiotropic inflammatory outcomes [15]. Due to its complexity in expressional regulation of IL-6/IL-6R, there is a discrepancy in the effect of IL-6 and its downstream pathway, JAK-STAT3 on melanocyte activation. For example, IL-6 and JAK-STAT3 are involved in the suppression of melanin production, growth inhibition, and cell death of melanocytes in conditions like vitiligo and in vitro melanocyte culture [16,17]. Contrary to these, we demonstrated that even with similar pY705-STAT3 status between young and adult keratinocytes, STAT3 activation did not change melanin production and melanocyte viability. Considering this, it is possible that the JAK-STAT3 pathway in melanocytes via elevated sIL-6 from adult keratinocytes, but not from young keratinocytes, may interplay with other kinase pathways such as MAPK and PI3K, influencing downstream transcription and secretory factors and ultimately affects dendrite morphology [15,18]. Additionally, effects of IL-6 on melanocytes could be dose or IL-6/IL-6R ratio dependent manner where intracellular signaling contexts is critical to exhibit specific effects such as melanogenesis and dendrite and melanosome transfer. In fact, Hakozaki et al. demonstrated that non-toxic level of IL-6 protein stimulates dendrite length and melanosome transfer in vitro melanocytes and melanocyte–keratinocyte coculture, respectively. Additionally, they also showed that elevated expression of IL-6R in age spots [19]. It is also of note that IL-6 is upregulated in skin disorders such as psoriasis [20], suggesting that it may trigger disease states through different signaling pathways than those in vitiligo. In age spots, IL-6R may be involved in chronic inflammation in the epidermis, constitutively stimulating melanocyte dendricity and inducing excessive melanin transfer. To further understand the detailed mechanism of IL-6/sIL-6R, we are currently investigating secretory patterns including IL-6/IL-6R of young and adult keratinocytes with or without UVB exposures and interactive effects on melanocyte dynamics by integrative multi-omics and epigenetic analyses. Finally, it is noteworthy that recent studies have demonstrated that UVA and visible light also play a role in epidermal pigmentation [21]. Addressing age-dependent alterations in the responsiveness of the epidermis to UVB and other light frequencies is crucial to developing age-appropriate strategies for photoprotection and skin-brightening care.

## 3. Materials and Methods

### 3.1. Assessment of Age Spots in Human Cutaneous Specimens

With the approval of the ethical committee of the Shiseido MIRAI Technology Institute (approval No. C10845, Yokohama, Japan), commercial facial cutaneous specimens (CTIBiotech, Lyon, France) containing hyperpigmented spots were used to assess the histological features of age spots, such as rete ridges, hyperpigmentation in basal keratinocytes, and thickened epidermis [9]. To examine the expression patterns of IL-6R, selected samples (*n* = 5, with 2 males and 3 females; 79–91 years of age; average age: 85 years old) were embedded in paraffin and subjected to fluorescent immunostaining.

### 3.2. Cell Cultures

Healthy human epidermal keratinocytes from neonatal Caucasian foreskin (Kurabo, Osaka, Japan, KK-4009), young Caucasian calf and African American abdominal skin (KK-4109, Kurabo, Osaka, Japan; 12 and 17 years old, respectively), and adult Caucasian abdominal skin (Kurabo, KK-4109; 48, 56, and 57 years old) were cultured in EpiLife medium (Thermo Fisher Scientific, Waltham, MA, USA) supplemented with 60 μM CaCl_2_ and HuMedia KG Growth Factor Kit (Kurabo). Healthy human epidermal melanocytes (Thermo Fisher Scientific) from a lightly pigmented neonatal Caucasian donor were cultured in Medium 254 (Thermo Fisher Scientific) supplemented with HMGS2 Growth Factor Kit (Kurabo). Keratinocytes and melanocytes were co-cultured with CnT-PRIME KM (CELLnTEC, Bern, Switzerland). All cultures were incubated at 37 °C in a 5% CO_2_ environment.

### 3.3. UVB Irradiation

The irradiance of the UV source was 0.20 mW/cm^2^, and the UV spectrum ranged from 280 nm to 320 nm. After examining a range of 10–200 mJ/cm^2^, we determined the optimal dose of UVB where keratinocytes did not exhibit significant DNA damage or cell death leading to a significant reduction in cell viability. In this condition, the cultured keratinocytes were irradiated with 10 mJ/cm_2_ UVB in a thermostatic UV irradiator (NK System, Osaka, Japan) 24 h after seeding and were collected either 2 or 4 days later for ELISA measurement.

### 3.4. Neutralization of the Function of Interleukin-6 Receptor (IL-6R) with Tocilizumab

An IL-6R-specific neutralizing antibody, Tocilizumab (NBP2-75192, Novus Biologicals, Centennial, CO, USA), was used to evaluate the function of IL-6R secreted by keratinocytes. Since we observed that adult keratinocytes were susceptible to Tocilizumab even at low concentration (50 pg/μL) upon viability assessment, we utilized young keratinocytes (Y0 or Y12) for Tocilizumab treatment at various doses, 50 pg/μL, 100 pg/μL, 250 pg/μL, and 500 pg/μL, for determining the optimal concentration. After the optimization procedure, either 250 pg/μL of Tocilizumab or mouse IgG1 control antibody (02-6100, Thermo Fisher Scientific) was added to the UVB-irradiated keratinocyte culture, incubated for 24 h, and then rinsed off with D-PBS(-) (Nacalai tesque, Kyoto, Japan) three times. The cultured melanocytes were seeded on either Tocilizumab or control antibody-treated keratinocytes in an initial culture ratio of keratinocytes to melanocytes of 5:1 and co-cultured in co-culture medium for 2 d.

### 3.5. ELISA Assay for Measurement of sIL-6R

The protein level of sIL-6R in the keratinocyte culture after UVB irradiation was determined using a Human IL-6R ELISA kit (Proteintech, Rosemont, IL, USA) according to the manufacturer’s instructions. The total protein amount of sIL-6R was normalized against the total number of viable keratinocytes. For viable cell counting, the fluorescence of either the alamarBlue cell viability reagent (Thermo Fisher Scientific) or Hoechst 33258 (Sigma-Aldrich, Burlington, MA, USA) was measured using a multi-detection microplate reader (POWERSCAN HT, BioTek, Agilent Technologies, Santa Clara, CA, USA). The melanin content was quantified by measuring the intensity of pigment in individual melanocytes in bright-field images from which the intensity of the surrounding background had been subtracted.

### 3.6. Fluorescent Immunostaining of Cultured Cells and Skin Tissues

The fluorescent immunostaining procedure for cultured cells was performed as previously described [9]. Briefly, fixed cultured cells and paraffin sections of cutaneous tissues were washed three times with PBST (1× PBS containing 0.1% Triton™ X-100 (Nacalai Tesque, Kyoto, Japan)) for 5 min each. The samples were subsequently blocked with 2.5% goat serum in PBST for one hour at room temperature and then incubated with appropriate primary antibodies (1:200 dilution for all primary antibodies in 1% goat serum with PBST) overnight. The following day, the samples were incubated with secondary antibodies against the respective species at the same dilution rate as the primary antibodies for 1.5 h at room temperature. The nuclei were counterstained with 4′,6-diamidino-2-phenylindole (DAPI) (H-1200; Vector Laboratories, Burlingame, CA, USA) for 10 min at room temperature. Static images of the immunostained samples were retrieved using confocal laser scanning microscopy with a 20× objective (LSM700, Zeiss, Oberkochen, Germany) and further analyzed using ImageJ (ver. 1.54p) [22]. Primary antibodies against the following were used: anti-human IL-6R (#MAB2271, R&D Systems, Minneapolis, MN, USA), anti-Melan-A (ab210546, Abcam, Cambridge, MA, USA), and anti-STAT3 (phospho-Y705) (ab76315, Abcam). The secondary antibodies used were Alexa Fluor™ 488-conjugated anti-rabbit IgG (A21206, Thermo Fisher Scientific) and Alexa Fluor™ 647-conjugated anti-mouse (A21236, Thermo Fisher Scientific). Melanin deposition and IL-6R expression were visualized using confocal microscopy (LSM700) with differential interference contrast (DIC).

### 3.7. Statistical Analyses

The imaging data for the statistical analyses were measured using ImageJ [22] and are presented as means ± standard deviations. The statistical analyses were performed with Prism 9 (Ver. 9.4.1, GraphPad, Boston, MA, USA), using Student’s *t*-test (paired, two-tailed) or multiple comparisons (one-way or two-way ANOVA) based on the experimental data. Differences with a value of *p* < 0.05 were considered statistically significant.

## 4. Conclusions

The study concludes that the age- and UVB-dependent upregulation of sIL-6R in keratinocytes plays a critical role in stimulating melanocyte dendricity. The findings suggest that IL-6R could be a key factor in the molecular mechanisms linking microinflammation to melanocyte activation and hyperpigmentation in age spots.

## Figures and Tables

**Figure 1 ijms-26-10971-f001:**
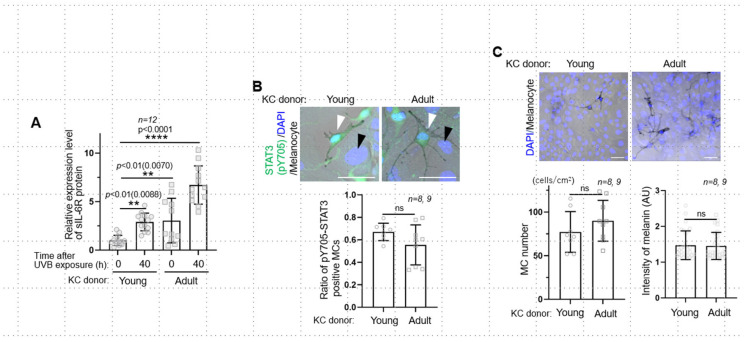
UVB-/age-dependent upregulation of sIL-6R secreted by keratinocytes and its effects on melanocytes. (**A**): Relative expression levels of sIL-6R protein in keratinocyte culture before and after 10 mJ/cm^2^ UVB irradiation. Note that the sIL-6 level in both young and adult keratinocytes was upregulated in a UVB exposure-dependent manner. Additionally, adult keratinocytes upregulated sIL-6R release more than young keratinocytes. (**B**): Activation of the JAK-STAT3 pathway in melanocytes (white arrowheads) but not in keratinocytes (black arrowheads) in the co-culture of young or adult UVB-irradiated keratinocytes with normal melanocytes. Nuclear-localized phospho-Tyr705 of STAT3 (pY705) in melanocytes showed no obvious difference between young and adult keratinocyte co-cultures. (**C**): Melanocytes co-cultured with both young and adult keratinocytes showed no significant changes in the number of melanocytes and melanin content. Scale bars: 50 μm. The data are represented as means ± SDs. KC, keratinocyte; ns: not significant.

**Figure 2 ijms-26-10971-f002:**
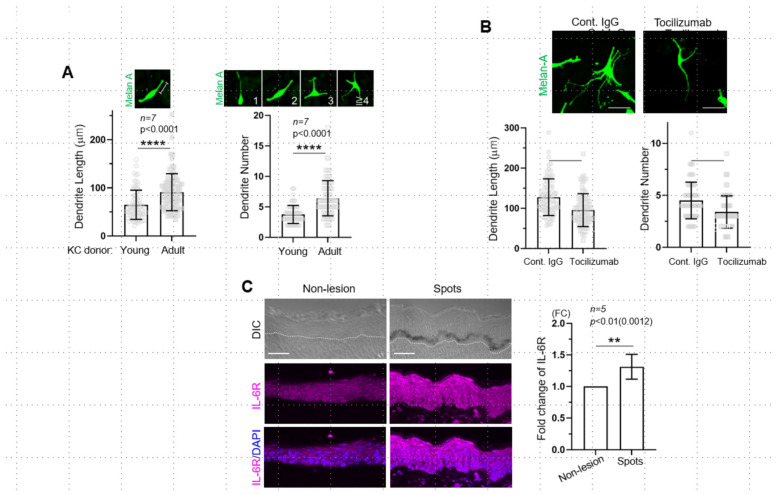
Stimulation of melanocyte dendricity mediated by sIL-6R and upregulated expression of IL-6R in age spots. (**A**): In the co-culture of either young or adult keratinocytes, the length and the number of melanocyte dendrites were significantly increased in an age-dependent manner. (**B**): The increase in both dendrite length and number was significantly suppressed in the co-culture of Tocilizumab-treated keratinocytes compared with those in the control IgG-treated keratinocyte co-culture (Cont. IgG). (**C**): Representative fluorescent images of IL-6R protein expression in the epidermis of non-lesions and age spots (*n = 5* each). The relative level of IL-6R protein was significantly upregulated in age spots, being ~1.2-fold higher than that in the non-lesion area. FC, fold change. Dashed lines denote dermal–epidermal junction. DIC, differential interference contrast. Scale bars: 50 μm (**B**,**C**). The data are represented as means ± SDs. KC, keratinocyte.

## Data Availability

Raw data were generated at MIRAI technology institute, Shiseido Co. Ltd. The data supporting the findings of this study are available from the corresponding author DI on request.

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
