# Peer review of "UVB-/Age-Dependent Upregulation of Inflammatory Factor Interleukin-6 Receptor (IL-6R) in Keratinocytes Stimulates Melanocyte Dendricity"

_ijms, 2025, doi:10.3390/ijms262210971_

Round 1
Reviewer 1 Report
Comments and Suggestions for Authors
The manuscript describes the role of IL-6R on melanocyte dendricity and the potential role in age spots (lentigo solaris). While the data presented are interesting, they are not complete to support the conclusions and the title is misleading.
The authors conclude the higher IL-6R production by keratinocytes from older donors is age-related, however, it might also be related to chronic sun exposure over decades. age spots appear in sun-exposed skin areas of elderly individuls not in sun-protected areas. No information is given on the origin of the keratinocytes, are they from biopsies from sun-exposed or sun-protected areas? Would be intersting to see if keratinocytes from sun-protected areas of older donors react more like keratinocytes from younger donore or more like keratinocytes from sun-exposed areas of the elderly.
Measuring IL-6R is not sufficient to analyze the described effects. Since IL-6R is the receptor for IL-6, it is important to measure also the levels of IL-6 in the cell cultures as well. The ratio of IL-6/IL-6R would provide important additional information. Are the effects on melanocyte dendricity due to IL-6R itself or rather due to reduction of IL-6 levels available to melanocytes? Here, addition of IL-6 to the cell cultures would provide additional information
While the UV-irradiation experiments are interesting, it needs to be described in more detail. The dose 10 mJ/cm2 UVB seems to be rather low, what was the irradiance of the UV-source, the spectrum of the radiation?
Comments on the Quality of English LanguageThe English language should be improved.
Author Response
|
Reviewer#1 Comments 1: The manuscript describes the role of IL-6R on melanocyte dendricity and the potential role in age spots (lentigo solaris). While the data presented are interesting, they are not complete to support the conclusions and the title is misleading. |
|
Response 1: We agree with that comment and changed the title to “UVB-/age-dependent upregulation of inflammatory factor interleukin-6 receptor (IL-6R) in keratinocytes stimulates melanocyte dendricity”. Please see Page1, lines 3-4. |
|
Reviewer#1 Comments 2: The authors conclude the higher IL-6R production by keratinocytes from older donors is age-related, however, it might also be related to chronic sun exposure over decades. age spots appear in sun-exposed skin areas of elderly individuls not in sun-protected areas. No information is given on the origin of the keratinocytes, are they from biopsies from sun-exposed or sun-protected areas? Would be intersting to see if keratinocytes from sun-protected areas of older donors react more like keratinocytes from younger donore or more like keratinocytes from sun-exposed areas of the elderly. |
|
Response 2: Reviewer #1 commented on whether the results might reflect both chronic photoaging and intrinsic aging. This is a very important point, and I appreciate the feedback as it is crucial for conveying the main focus of this paper. In our study, we used keratinocytes from non-exposed areas (abdomen and calf) and performed a single UVB irradiation. Therefore, the results are not influenced by chronic photoaging, which was a concern of Reviewer #1. We have specified the details of the keratinocyte origin in the Materials and Methods section. Please see page 10, lines 173-176. |
|
Reviewer#1 Comments 3: Measuring IL-6R is not sufficient to analyze the described effects. Since IL-6R is the receptor for IL-6, it is important to measure also the levels of IL-6 in the cell cultures as well. The ratio of IL-6/IL-6R would provide important additional information. Are the effects on melanocyte dendricity due to IL-6R itself or rather due to reduction of IL-6 levels available to melanocytes? Here, addition of IL-6 to the cell cultures would provide additional information. |
|
Response 3: Thank you for pointing out crucial consideration regarding IL-6 level. As shown in the preliminary results in Figure 1, the expression of IL-6 protein in neonatal keratinocytes increased about twofold 6 hours after UVB irradiation (10 mJ/cm2). The result is already known as acute phase inflammatory response of IL-6 upregulation [1]. On the other hand, the expression of IL-6R protein did not significantly increase within 6 hours of irradiation but rather delayed its upregulated expression 40h after irradiation (please see Fig.1A in the manuscript). These results imply that the expression ratio of IL-6 and IL-6R together with their expression patterns through short to long-term time window would be complex and critical. As the reviewer suggested, from our ongoing research, it has become clear that the timing of expression and half-life of both IL-6 and IL-6R at the protein level need to be considered for understanding underlying mechanism of IL-6/sIL-6R on melanocyte dendricity. Furthermore, in the experiment proposed by the reviewer where IL-6 is externally added, careful examination is required to determine the optimal concentration of recombinant human IL-6 protein including its half-life as well as the timing of addition for the downstream experiment, keratinocyte-melanocyte coculture. Therefore, these lines of experiments regarding IL-6/IL-6R ratio will be addressed for our ongoing research by utilizing multi-omics and epigenetic analyses. We have added new references and included a discussion emphasizing the importance of considering the expression ratio, timing, and half-life of IL-6/IL-6R in future research, as well as outlining the prospects of our ongoing studies. Please see red sentences in pages 9 and 10.
[1] Hiromi, N.; Nakano, H.; Matsuzaki, Y. et al. Immunohistochemical analysis of in vivo UVB-induced secretion of IL-1alpha and IL-6 in keratinocytes. Mol. Med. Rep. 2011, 4, 611-614.
|
|
Reviewer#1 Comments 4: While the UV-irradiation experiments are interesting, it needs to be described in more detail. The dose 10 mJ/cm2 UVB seems to be rather low, what was the irradiance of the UV-source, the spectrum of the radiation? |
|
Response 4: We appreciate the comment. The irradiance of the UV source was 0.20 mW/cm², and the UV spectrum ranged from 280 nm to 320 nm, which is also described in the Materials and Methods section. The reason for setting the irradiance to 10 mJ/cm² is based on the following two points: (1) After examining a range of 10–200 mJ/cm², we determined the optimal condition where keratinocytes did not exhibit significant DNA damage or cell death leading to a reduction in cell viability. Additionally, (2) considering the physiological conditions of Japanese skin, the UVB irradiation dose during midday in summer that causes erythema is approximately 20 to 60 mJ/cm² (15–50 minutes, 1 MED [minimal erythema dose]) [1]. Considering the above two points and the situation where keratinocytes are directly exposed to UVB in vitro, we decided on that UVB dose. Please see lines 185-188 in page 11.
[1] Ichihashi, M.; Ando, H. The maximal cumulative solar UVB dose allowed to maintain healthy and young skin and prevent premature photoaging. Exp Dermatol. 2014, 23, 43-46. |
|
Reviewer#1 Comments 5: The English language should be improved. |
|
Response 5: I also appreciate the comment. We have also revised and proofread the manuscript. |
|
Reviewer#2 Comments 1: However, the study results showed that despite the higher secretion of sIL-6R in old keratinocytes compared to young keratinocytes, STAT3 activation in melanocytes remained unchanged. Furthermore, it was described that sIL-6R did not change melanin production, but rather induced an increase in dendrite number and length, thereby influencing age spot formation. Although the authors expressed this fact in the discussion, it is contrary to generally recognized research results. That is, when sIL-6R is abundant, there is a high possibility that JAK-STAT3 activity will increase, and since IL-6 signaling is known to suppress melanogenesis, the author does describe this, but the explanation or consideration seems insufficient. If the author could refer to previous studies on this part and suggest a possible mechanism among other mechanisms that could induce conflicting results, it would be able to provide more insight for future research. |
|
Response 6: Thank you for pointing out the contradictions regarding the effects of IL-6 on melanocytes and the downstream activation of STAT3. First, regarding no significant difference in pY705-STAT3 between young and adult keratinocytes, we speculate that the phosphorylation level may have reached a plateau. Considering this, it is possible that the JAK-STAT3 pathway in melanocytes activated via elevated sIL-6R from adult keratinocytes, but not from young keratinocytes, in conjunction with other kinase pathways such as MAPK and PI3K, influences downstream transcription and secretory factors and ultimately affects dendrite morphology [1][2]. Additionally, effects of IL-6 on melanocytes could be dose or IL-6/IL-6R ratio dependent manner where intracellular signaling contexts is critical to exhibit specific effects such as melanogenesis and dendrite and melanosome transfer. In fact, Hakozaki et al. demonstrated that at non-toxic level of IL-6 protein promotes dendrite length and melanosome transfer in vitro melanocyte and melanocyte-keratinocyte cocultures, respectively. On the other hand, IL-6 is well known to be involved in the suppression of melanin production, growth inhibition, and cell death of melanocytes in conditions like vitiligo and in vitro [2][3]. However, it is also known that IL-6 is upregulated in skin disorders such as psoriasis [4], suggesting that it may trigger disease states through different signaling pathways than those in vitiligo. As we responded to reviewer #1's comments 3, IL-6 and sIL-6R have different expression timings and protein half-lives, and the various downstream activation states and activation timings influenced by the IL-6/sIL-6R ratio in classical, trans, and cluster signaling have been noted in diverse inflammatory diseases involving IL-6/sIL-6R [1]. However, the detailed mechanisms remain largely unknown. We speculate that the spatiotemporal control of IL-6/sIL-6R is crucial for dendrite formation and melanin production in melanocytes, and we are currently advancing our research to elucidate the detailed mechanisms. We have added new references and these considerations in the discussion section. Please see red sentences in pages 9 and 10.
[1] Schmertl, T.; Lokau, J.; Garbers, C. IL-6 Signaling in Immunopathology: From Basic Biology to Selective Therapeutic Intervention. Immunotargets. Ther. 2025, 14, 681-695. [2] Ihara, S.; Nakajima, K.; Fukada, T. et al. Dual control of neurite outgrowth by STAT3 and MAP kinase in PC12 cells stimulated with interleukin-6. EMBO J. 1997, 16, 5345-5352.
[3] Hakozaki, T.; Wang, J.; Laughlin, T. et al. Role of interleukin-6 and endothelin-1 receptors in enhanced melanocyte dendricity of facial spots and suppression of their ligands by niacinamide and tranexamic acid. J. Eur. Acad. Dermatol. Venereol. 2024, 38, 3-10.
[4] Balato, N.; Di Costanzo L.; Balato, A. et al. Psoriasis and melanocytic naevi: does the first confer a protective role against melanocyte progression to naevi? Br. J. Dermatol. 2011, 164, 1262-1270.
|
|
Specific comments: We changed the label of figures from “Old” to “Adult” according to the proofread of the manuscript. |

Reviewer 2 Report
Comments and Suggestions for Authors
The authors consider this manuscript a meaningful one, elucidating the increase in melanocyte dendrite number and length by sIL-6R. However, the study results showed that despite the higher secretion of sIL-6R in old keratinocytes compared to young keratinocytes, STAT3 activation in melanocytes remained unchanged. Furthermore, it was described that sIL-6R did not change melanin production, but rather induced an increase in dendrite number and length, thereby influencing age spot formation. Although the authors expressed this fact in the discussion, it is contrary to generally recognized research results. That is, when sIL-6R is abundant, there is a high possibility that JAK-STAT3 activity will increase, and since IL-6 signaling is known to suppress melanogenesis, the author does describe this, but the explanation or consideration seems insufficient. If the author could refer to previous studies on this part and suggest a possible mechanism among other mechanisms that could induce conflicting results, it would be able to provide more insight for future research.
Author Response

(The authors gave the same response as above.)
